# Movement Sensing Opportunities for Monitoring Dynamic Cognitive States

**DOI:** 10.3390/s24237530

**Published:** 2024-11-25

**Authors:** Tad T. Brunyé, James McIntyre, Gregory I. Hughes, Eric L. Miller

**Affiliations:** 1U.S. Army DEVCOM Soldier Center, Natick, MA 01760, USA; gregory.hughes@tufts.edu; 2Center for Applied Brain and Cognitive Sciences, Tufts University, Medford, MA 02155, USA; james.mcintyre659218@tufts.edu (J.M.); eric.miller@tufts.edu (E.L.M.); 3Department of Electrical and Computer Engineering, Tufts University, Medford, MA 02155, USA

**Keywords:** inertial measurement units, optical motion capture, cognitive state estimation, workload, uncertainty, machine learning, movement dynamics

## Abstract

In occupational domains such as sports, healthcare, driving, and military, both individuals and small groups are expected to perform challenging tasks under adverse conditions that induce transient cognitive states such as stress, workload, and uncertainty. Wearable and standoff 6DOF sensing technologies are advancing rapidly, including increasingly miniaturized yet robust inertial measurement units (IMUs) and portable marker-less infrared optical motion tracking. These sensing technologies may offer opportunities to track overt physical behavior and classify cognitive states relevant to human performance in diverse human–machine domains. We describe progress in research attempting to distinguish cognitive states by tracking movement behavior in both individuals and small groups, examining potential applications in sports, healthcare, driving, and the military. In the context of military training and operations, there are no generally accepted methods for classifying transient mental states such as uncertainty from movement-related data, despite its importance for shaping decision-making and behavior. To fill this gap, an example data set is presented including optical motion capture of rifle trajectories during a dynamic marksmanship task that elicits variable uncertainty; using machine learning, we demonstrate that features of weapon trajectories capturing the complexity of motion are valuable for classifying low versus high uncertainty states. We argue that leveraging metrics of human movement behavior reveals opportunities to complement relatively costly and less portable neurophysiological sensing technologies and enables domain-specific human–machine interfaces to support a wide range of cognitive functions.

## 1. Introduction

Whether in the form of subtle finger movements or whole-body coordinated activity, humans are constantly moving. Research suggests that movement results from, indicates, and guides perception and thought, forming the basis of cognitive science theories of embodied cognition and perception-action feedback loops [1,2,3,4]. The notion that subtle alterations in movement behavior can indicate cognitive states is readily exemplified through everyday experience: a shaking knee can indicate stress and anxiety, erratic head movements can indicate confusion or disorientation, increased postural stability can indicate engagement in a task, and anterior trunk lean can indicate mental workload [5,6,7,8,9]. In other words, humans embody the inherent mental demands of a context or task and produce measurable behavioral traces of this embodiment that can be leveraged to peer inside the “black box” of cognition.

Prior research examining behavioral or neurophysiological correlates of cognitive states is largely restricted to laboratory contexts with highly controlled environments and tasks, leveraging diverse multi-modal sensors. For example, to assess mental workload, researchers have variably measured pupil diameter and patterns of eye movements using infrared eye tracking [10,11,12], heart rate and heart rate variability using physiological monitoring [13,14,15], prefrontal cortical brain activity using functional near-infrared spectroscopy [16,17], frequency-domain oscillatory brain activity using electroencephalography (EEG) [18,19], acoustic modulations of speech [20], and/or subjective assessments such as the NASA task load index (NASA-TLX) [21]. While each of these measures offers promise within specific contexts, they are also subject to several inherent limitations. Indeed, neurophysiological sensing is largely restricted to non-ambulatory contexts due to low signal-to-noise ratios and motion artifact, speech analysis is restricted to tasks involving spoken language, and subjective instruments often necessitate disrupting task engagement to probe self-reported states. Movement sensing offers an unintrusive supplement or alternative to traditional neurophysiological sensors used in the cognitive and brain sciences, with emerging research suggesting that it can be leveraged for cognitive state estimation.

## 2. Movement Sensing and Cognitive State Estimation

Human movement sensing is mostly performed in one of two primary modalities, using optical motion capture (OMC) or inertial measurement units (IMUs). OMC uses a synchronized array of infrared cameras to track sequences of Cartesian coordinates as markers (e.g., retroreflective marker balls) move through a limited measurement volume. In systems using markers, passive or active markers are typically placed over anatomical landmarks (e.g., knee, sternum, clavicle) or on tools (e.g., on a rifle). Based on sequences of translational and rotational marker movement, an image acquisition system processes and visualizes movement patterns of tracked objects and derives quantitative movement metrics of kinematics (e.g., velocity, acceleration) and/or kinetics (e.g., force, power) [22]. OMC has very high precision, with overall precision typically below 100–200 µm depending upon camera capabilities, tracking volume, calibration procedures, and the position of trackable objects [23,24]. However, OMC hardware and software are costly and resource intensive for training, operation, and maintenance [25]. They also track over a restricted volume of space, reducing applicability to field environments.

In contrast, IMU-based sensors use an array of gyroscopes, accelerometers, and/or magnetometers to track an object’s acceleration, velocity, rotation (roll, pitch, yaw), and/or heading relative to a global reference frame [26]. The gyroscope provides information regarding angular rate, the accelerometer provides information regarding force and acceleration, and the magnetometer (when equipped) provides information regarding the local magnetic field. Many modern handheld and wearable devices use integrated IMUs including smartphones, tablets, and fitness trackers, and they are becoming increasingly miniaturized while maintaining reasonable accuracy and precision [27]. While IMUs cannot achieve the precision of OMC, particularly over extended periods of time (i.e., as drift accumulates), they present sensing opportunities with relatively unintrusive, portable, low cost, and flexible technologies that can be used in several applications: seated or ambulatory, stationary or moving, and with individuals or groups [25,28,29,30].

We consider four domains in which research may benefit from movement sensing, whether via OMC or IMU, to provide insights into transient cognitive states: sports, healthcare, driving and navigation, and military. Thereafter, we describe a new dataset and provide a detailed demonstration of how OMC-based movement sensing may provide insights into dynamic mental states of uncertainty while military personnel move a weapon during a simulated marksmanship task.

### 2.1. Sports

Movement sensing is increasingly popular in sports medicine, with an emphasis on tracking and optimizing training trajectories and providing a basis for data-driven feedback between players and coaches [31,32]. While most work examining movement sensing in athletes tends to emphasize injury and fatigue prevention [33,34,35], a few recent studies have suggested that movement sensing can be used to predict athlete intentions (a high-level cognitive function involving goal-setting, decision-making, and planning) immediately prior to overt movement.

In one study, participants performed a series of trials where they would stand and then transition to run in one of eight directions (e.g., straight ahead, to the right, to the left) based on a visual cue [36]. Participants’ movement was tracked using a combination of OMC, IMUs, force plates, and electromyography (EMG) to capture muscle activation and movement kinetics and kinematics. Using a machine learning algorithm (XGBoost) with ensemble learning and gradient boosting, the authors were able to identify participants’ intent to move, and in which direction, approximately 100 ms before the onset of detectable velocity change at the center of mass (i.e., absolute kinematic start). In other words, the algorithm can detect two mental states of interest: preparation to execute a movement, and an intent to move in a particular direction. The authors suggest that kinetics measured with OMC provided the most valuable contribution to the algorithm, perhaps given its relative precision.

In sports, the ability to detect movement intent can provide insight into when an individual recognizes and successfully interprets game dynamics and uses them to guide their own behavior; it can also be valuable for predicting offensive and defensive plays in team sports contexts.

### 2.2. Healthcare

In healthcare, movement sensing has been used to infer mental states of both physicians and patients. With physicians, tracking the movement of surgical tools during surgery has been associated with varied skill level, with relatively novice and uncertain (a mental state of limited knowledge or information, making it difficult to predict outcomes or make decisions) surgeons showing different movement patterns relative to expert surgeons [37]. For example, Cao and colleagues tracked laparoscopic tool use during four surgical tasks (dissection, suturing, knot tying, suture cutting) and found distinct movement features, including velocity, were related to surgeons’ uncertainty of position and orientation within the body cavity [38]. Furthermore, when pathologists examined (i.e., zooming, panning) tissue biopsies, several movement-related features including movement entropy were related to diagnostic accuracy in detecting cancerous lesions, suggesting that physicians with relatively high certainty and confidence show relatively predictable (i.e., less erratic) movements [39]. In a study with surgeons performing arthroscopic surgery, machine learning classifiers were able to identify confusion states with over 94% accuracy by examining head and eye movements alone [40]. In medical training, algorithms resulting from this work could be used to automate the detection of uncertainty states and trigger expert remediation, AI-enabled assistance, and recommender systems.

With patients, tracking hand movements during cognitive tasks can differentiate those with versus without autism spectrum disorder (ASD) or major depressive disorder [41,42], suggesting the value of limb movement tracking to detect psychological conditions. In age-related clinical disorders, insole balance sensors have been used to predict mild cognitive impairment based on balance-related features [43], and features of gait measured using OMC can predict dementia onset [44] and mild cognitive impairment [45]. Head movement tracking has also been shown sensitive to chronically high mental workload levels that might be associated with changes in well-being and health [46].

### 2.3. Driving and Navigation

Identifying reliable markers of driver inattention, disorientation, or workload could prove valuable for human–machine integration with semi-autonomous driving systems. While a lot of research has considered the value of oculomotor metrics such as eye gaze and blink frequency and duration, some work has considered relatively gross movement dynamics of drivers. For example, using sensors affixed to steering wheels in driving simulators, researchers have found that rotational entropy of steering wheel movements is indicative of various mental states including workload and inattention [47,48].

Similarly, sensing the entropy of head movements during navigation tasks can be indicative of disorientation and uncertainty [7]. Specifically, when yaw-based head movements are relatively unpredictable and erratic, navigators are more likely to be experiencing uncertainty regarding direction of travel. Interestingly, entropy is a recurring feature in earlier work, suggesting that the inherent unpredictability and disorder of movement behavior can indicate mental states of confusion or uncertainty. Interestingly, when people are placed under conditions of stress and uncertainty, neuroscientists have found increased entropy in both functional brain activity and heart rate dynamics [49,50,51,52]; it could be the case that entropy-related movement features reflect these neural dynamics.

### 2.4. Military

In military contexts, dynamic coordinated body movements are fundamental to many common tasks. Whether maintaining a vehicle, hiking a mountain, entering a building, or setting up a roadblock, military personnel must coordinate their body and equipment while variably experiencing states of stress, uncertainty, and workload.

In the case of rifle marksmanship, military personnel constantly observe their environment, orient towards potential threats, make decisions regarding the posture and intent of potential threats, and then act accordingly [53,54]. While research in sports, healthcare, and driving and navigation domains suggest that movement dynamics might indeed indicate mental states, to our knowledge, no studies have examined whether rifle movement dynamics during marksmanship tasks might be indicative of unfolding mental states of uncertainty, workload, or stress.

To fill this gap, we present data collected from military personnel engaged in a simulated marksmanship task while we used OMC sensors to track rifle position and compute features of movement dynamics. In the next section, we describe the design, analysis, and results of this analysis.

## 3. Classifying Uncertainty States via Rifle Movement Dynamics

As part of a larger study previously reported [55,56], we used a marksmanship simulation system to collect data from military personnel engaged in a shoot/don’t-shoot decision-making task. For these analyses, we focused on 6DOF OMC data of rifle movements collected during discrete task trials to assess whether time series patterns of movement data might reliably indicate self-reported uncertainty states.

### 3.1. Participants, Design and Procedure

A total of 83 male (Mean age 23.1) military personnel (active-duty U.S. Army soldiers) voluntarily provided written informed consent to participate in a study approved by the institutional review boards at Tufts University and the U.S. Army Combat Capabilities Command Armaments Center. During a series of laboratory sessions, participants completed several tasks measuring cognitive, affective, physiological, and biochemical responses to acute stress. To measure cognitive responses, tasks included spatial orienting, recognition memory, and simulated marksmanship.

For the purpose of this analysis, we focus only on the simulated marksmanship task the details of which are described below. This task involved two phases: learning and testing. During learning, participants learned how to distinguish a friendly versus enemy camouflage pattern to an accuracy criterion of 80%. During testing, these camouflage patterns were placed on simulated avatars with systematically varied clarity, similar to a traditional perceptual decision-making task [57,58]. Specifically, camouflage visual clarity was varied across six conditions that overlaid one pattern over the other to increase confusability: 100% pattern A and 0% pattern B, 60–65% pattern A and 35–40% pattern B, and 51% pattern A and 49% pattern B, and vice versa. In this manner, any given avatar was relatively easy or difficult (and in some cases impossible) to distinguish as friendly versus enemy. This manipulation was intended to ensure that participants would experience variable subjective uncertainty levels over the course of trials.

The camouflaged avatars were used in a virtual reality scenario (built using the Unity3d (version 2018.2.6f1) engine [59]) that simulated an avatar walking towards the participant in the virtual world. Over the course of 15 trials, the participant had to decide whether the avatar was friendly or enemy (based solely on their camouflage pattern) and therefore whether to allow the avatar to pass (friendly) or to shoot (enemy) the avatar using a gaming rifle. At the outset of each trial, the participant was instructed to return the rifle to the downward-facing (i.e., low-ready) position; if they decided to engage an enemy, they would shoulder the rifle, aim at the avatar, and pull the trigger. If they decided to let a friendly pass, they would leave the rifle in the downward-facing position and push a small button on the barrel of the rifle. Critically, immediately after each shooting decision, participants would rate their certainty on a scale from 1 (very low) to 6 (very high). The task was performed under stress, with participants receiving a mild electric shock to the torso when they made an incorrect decision (i.e., shooting a friendly, or missing an enemy); shock was administered only after they made their certainty rating.

The simulated marksmanship task was performed two to three times by each participant (on separate days) in a large-scale projection screen-based cave automatic virtual environment (CAVE). The rifle was equipped with an array of retroreflective marker balls that were registered and tracked by a TRACKPACK/E infrared motion tracking system (Advanced Realtime Tracking, GmbH, Weilheim, Germany). This afforded sensing of 6DOF (*xyz*, roll, pitch, yaw) rifle movement at 60 Hz, recorded using the DTrack (version 3.1.1) software (also by Advanced Realtime Tracking).

### 3.2. Data Processing

Data were collected from a total of 3735 trials, with the rifle being raised and a trigger pull occurring on approximately half (1875) of all trials; these trials were carried forward for pre-processing movement trajectories. Note that we did not include in the analysis trials where the participant let the avatar pass as there was no appreciable movement of the rifle in such cases.

For the 1875 trials of interest, time series data from the OMC sensors were time-locked to the onset of a trial, when an avatar first became perceptible in the virtual scene, and the end of a trial, when the trigger was pulled to fire upon the avatar. Upon initial inspection of our data, we found that when participants moved their rifle from the low-ready to a shouldered position, movement was primarily within the translational degrees of freedom (i.e., *xyz* axes; predominately upward), with less movement in the rotational degrees of freedom (roll, pitch, yaw). Thus, analyses are restricted to the former. Example rifle movement trajectories are depicted in Figure 1.

Pre-processing resulted in the removal of 250 trials due to extremely brief (i.e., <5 s) or extremely long (i.e., >1 min) trajectories typically due to either tracking error (the former) or scenario software errors (the latter).

To derive features from the remaining 1625 trajectories, we leveraged two Python packages: the Time Series Feature Extraction Library (TSFEL [60]), and the Nonlinear Analysis Core’s NONANLibrary [61]. Together, the TSFEL and NONANLibrary packages allow for the calculation of features in the temporal, probability, spectral, divergence, and fractal domains. A full list of features can be found at the TSFEL and NONANLibrary GitHub pages. A total of 385 features for velocity (V) and each of the X, Y, and Z translational axes were computed resulting in 1540 features per trajectory. Relative to the participant’s perspective, the Y axis reflects superior–inferior (up/down) movement of the rifle, the X axis reflects medial–lateral (left/right) movement of the weapon, and the Z axis reflects anterior–posterior (forward/backward) movement of the weapon.

Data analysis was intended to assess whether the movement dynamic features could classify low versus high certainty states. A given trial for a given participant was deemed low certainty if the score was below the median score across all trials for that individual, or high certainty if the score was at or above the median. This process resulted in a total of 581 trials included in the low certainty class, and 1044 trials included in the high certainty class.

### 3.3. Data Analysis

To evaluate whether the extracted features can predict (un)certainty, we conducted a five-fold cross-validation, utilizing 80% of the data for training and 20% for testing in each fold. Feature selection was performed on the training set using the Terminating Random Experiments Selector (T-Rex Selector) [62]. The T-Rex Selector is notable for its use of dummy variables to control the false discovery rate (FDR), ensuring that the proportion of falsely identified variables among all selected variables meets a user-defined criterion which was set at 0.05 for all analysis in this paper.

The selected features from the training set were then used to train a scikit-learn (version 1.4.0) pipeline, which included a robust scaler and a logistic regression model. Predictions were made using the selected features and the trained pipeline. The predictive models achieved an average F1 score of 0.77, indicating that the rifle trajectory features are effective in predicting shot certainty (accuracy 0.66, misclassification rate 0.34, precision 0.67, recall 0.92, MSE 0.34), detailed in Table 1.

### 3.4. Results

The most frequently selected features across the five iterations are detailed in Table 2. Notably, selected features including Lyapunov exponents [63] and the slope of the power spectrum for the magnitude of the velocity, were related to the predictability or regularity of the trajectories in space. Two features were selected in most of the five iterations. First, the Lyapunov exponent from the velocity magnitude was selected in all five iterations. This feature had a negative weight in each model iteration, suggesting that decreased regularity (i.e., higher Lyapunov exponent) in trajectory velocity is associated with lower certainty. Second, the Lyapunov exponent of the lateral (X) position, was selected in four out of five iterations and also had a negative weight, indicating that predictability in the participant’s medial–lateral rifle movement (X axis) is related to lower certainty.

To test the statistical difference in the two most frequently selected features, we ran two-sample Kolmogerov Smirnov tests using Scipy’s (version 1.10.1) kstest method. Both features achieve a significant *p*-value (*p* < 1 × 10^−5^), suggesting the samples come from different distributions (Figure 2).

## 4. Discussion

Analysis of this data set demonstrated that features derived from rifle movement dynamics, particularly those related to the trajectories’ spatial predictability/regularity, can effectively distinguish between low and high uncertainty states in a simulated marksmanship task. Specifically, the Lyapunov exponents of both velocity magnitude and lateral (X-axis) movement consistently emerged as key predictors, with higher variability (i.e., lower predictability) in these movement patterns associated with lower certainty. This finding aligns with previous research in healthcare [38,39], navigation [7], and driving [47,48] contexts, which also suggests that increased movement irregularity or unpredictability (typically measured via entropy) can reflect heightened cognitive demand or uncertainty.

The consistent selection of features such as the Lyapunov exponent and velocity spectral slope highlights the utility of these and related movement characteristics in cognitive state estimation. The fact that movement dynamics could be predictive of cognitive uncertainty states, particularly in a highly stressful and dynamic task like marksmanship, supports the notion that movement sensing with OMS and/or IMUs can be an unobtrusive, real-time indicator of cognitive processes. Importantly, this extends previous research in domains like healthcare and navigation by demonstrating that rifle movement, specifically in military tasks, is a valuable proxy for underlying cognitive states.

These results suggest several practical implications. In military contexts, real-time detection of uncertainty based on movement features could be leveraged to provide adaptive feedback to trainees, optimize training protocols, quantify the transition from novice to expert, and reduce decision-making errors. There are also several promising civilian applications of our results. In law enforcement training, movement tracking for mental state classification it could improve decision-making under stress by providing real-time feedback on cognitive states like uncertainty, enabling tailored interventions that enhance officers’ confidence and resilience [64,65]. In sports training, movement tracking can optimize performance and injury prevention by revealing cognitive states related to focus, anticipation, and fatigue [66]. Within healthcare, this approach can aid in medical training by identifying moments of uncertainty in procedures, for example during surgical training [67]. For driver training and navigation, movement sensing could enhance safety by detecting fatigue or disorientation, encouraging adaptive guidance to reduce driver stress [68]. Finally, in workplace safety and efficiency, tracking subtle movement dynamics in high-risk jobs can inform training to manage workload and fatigue, improving ergonomics and reducing injury risks [69,70]. These applications demonstrate how movement sensing can support real-time, adaptive feedback across diverse high-demand environments. Across domains, classifying uncertainty states from movement behavior could also assist in quantifying the progression from novice to expert during training.

Future research should explore the potential for integrating movement-based cognitive state estimation with other physiological or environmental sensing modalities, as well as testing these methods in real-world training and operational contexts. Sensor fusion could improve the accuracy and precision of uncertainty state classification; however, it is compelling that we can achieve moderate-to-high model performance with a single sensor suitable for resource-constrained settings. It will also explore whether there are unique movement signatures related to similar but dissociable cognitive states such as cognitive workload, uncertainty, and acute stress. More broadly, our results suggest that human movement dynamics may prove valuable for classifying a wide range of neurocognitive states accompanying diverse work-related movements; while we focused on healthy, neurotypical participants, there are also potential applications for clinical surveillance and diagnosis.

## 5. Conclusions

In conclusion, we demonstrate that the movement dynamics characterizing weapon trajectories offer a promising avenue for estimating cognitive states such as uncertainty in complex, high-stress tasks. The findings pave the way for further exploration into how subtle changes in movement behavior can reveal cognitive states, offering new opportunities for sensing and cognitive state monitoring in both research and applied settings.

## Figures and Tables

**Figure 1 sensors-24-07530-f001:**
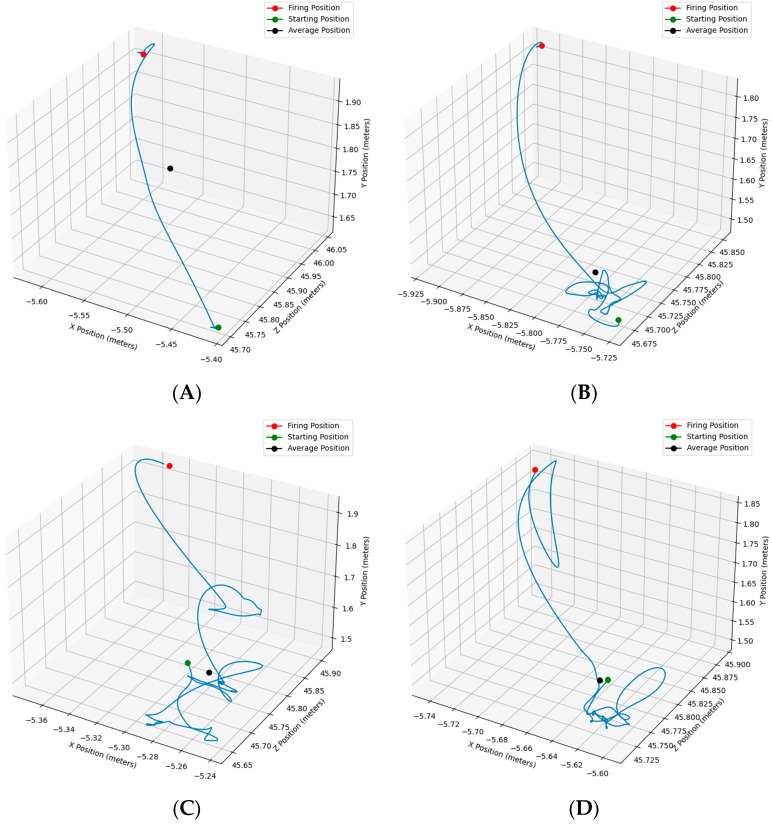
Example time-series rifle movement data, demonstrating either low variability panel (**A**), or high variability early (**B**), middle (**C**), and/or late (**D**) in the trajectory.

**Figure 2 sensors-24-07530-f002:**
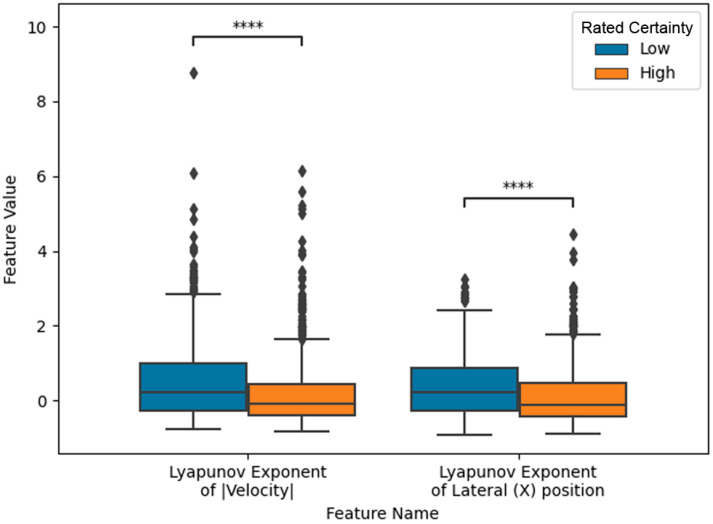
Box plot distinguishing low versus high rated uncertainty for two notable features, including 95% confidence intervals and an indication of pairwise statistical significance (**** *p* < 0.00001).

**Table 1 sensors-24-07530-t001:** Convergence of the model across the five iterations, including the number of features selected and five key performance variables.

Iteration	Number of Features	F1	Accuracy	Precision	Recall	MSE
1	11	0.775	0.664	0.682	0.897	0.336
2	8	0.763	0.645	0.671	0.883	0.355
3	6	0.766	0.651	0.677	0.883	0.348
4	11	0.768	0.661	0.684	0.877	0.339
5	7	0.775	0.657	0.670	0.920	0.343

**Table 2 sensors-24-07530-t002:** The 5 most frequently selected trajectory features across iterations of the 5-fold cross validation, mean feature weights, and a description of each feature. VM = velocity magnitude, AMI = average mutual information using Stergiou method.

Feature	Weight	Description
Lyapunov Exponent of VM	−0.29	The rate at which small differences in velocity grow over time, indicating sensitivity to initial conditions and chaos.
Lyapunov Exponent of X	−0.27	The rate at which small differences in X-axis (lateral) movement grow over time, indicating sensitivity to initial conditions and chaos.
Lyapunov Exponent of Y	−0.17	The rate at which small differences in Y-axis (vertical) movement grow over time, indicating sensitivity to initial conditions and chaos.
Spectral Slope	−0.16	Power of a trajectory’s velocity changing across different frequencies, revealing smoothness or complexity of the trajectory.
AMI (Stergiou) of X	−0.30	Nonlinear dependencies and predictability of X-axis movement over time, how much information past values provide about future values.

## Data Availability

The original data presented in the study are openly available at the Harvard Dataverse, persistent link: https://doi.org/10.7910/DVN/WZEHSR accessed on 23 September 2024.

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
