# Peer review of "Movement Sensing Opportunities for Monitoring Dynamic Cognitive States"

_sensors, 2024, doi:10.3390/s24237530_

Round 1

Reviewer 1 Report

Comments and Suggestions for Authors

The concept that motion detection using OMS and IMU can be an unobtrusive real-time indicator of cognitive processes has already been studied and proven in many fields and has reached the level of practical use.

However, due to the special limited environments of the military field, it is true that such research is very rare in stressful and dynamic tasks such as shooting, and we agree that related research is necessary.

In particular, it seems to be a very rare study that has proven its usability through real-time detection of uncertainty based on motion characteristics in a military context.

However, the methodology and measurement methods for related studies are already known, and it is not an original idea or method that can be specifically identified. It is acknowledged that the study was conducted in a field that was limited due to the special nature of the application field and its usability was proven through related results, but other than the fact that the application field could be expanded, there does not seem to be much other academic achievement.

If we want to increase the academic contribution, it would be better if the author integrates motion-based cognitive state estimation, which he suggested as a future research field, with other physiological or environmental detection modes to prove the usability of related results.

Author Response

Comment 1: If we want to increase the academic contribution, it would be better if the author integrates motion-based cognitive state estimation, which he suggested as a future research field, with other physiological or environmental detection modes to prove the usability of related results.

Response 1: We agree that this is an exciting future application – namely, multimodal sensor fusion with motion and physiology sensing. However, there are many contexts where it is not possible or resource efficient to track and fuse multiple data streams; herein, we demonstrate that a single data stream derived from motion capture can account for moderate-to-high levels of variance in uncertainty states. We believe this is a compelling result and highlights the potential for single-sensor state monitoring and classification, without the need for additional sensors. We now discuss this point in the revised manuscript.

Reviewer 2 Report

Comments and Suggestions for Authors

Authors performed feature extraction and classification of optical motion capture (OMC) data to relate the motion dynamics in virtual marksmanship task to cognitive demand or uncertainty. With the features selected, high classification rate of cognitive demand was achieved. The topic is of interest to the readers of this journal.

The manuscript needs improvement. Please see the following comments.

1) Abstract: a sentence to claim the problem or pain point of the related studies is necessary. The sentences in Line 164-168 can be rewritten, condensed and used to claim the problem.

2) Results: this is the section that needs most revision.

2.1) In a classification task, not only the performance of the classification, but also the cases of misclassification need to be analyzed. This analysis could guide you and readers to understand clearly the reason of the misclassification, the ambiguity of the dataset itself, or imperfect preprocessing or processing.

2.2) the details of iterations need to made clear. The number of features, features selected (even though frequently selected features have been shown in the manuscript), performance index values after each iteration, the convergence of the learning need to be shown.

3) Word usage: it is better to use “machine learning” to replace “deep learning”.

Author Response

Comment 1: Abstract: a sentence to claim the problem or pain point of the related studies is necessary.

Response 1: We now include a sentence in the abstract that makes this pain point more transparent.

Comment 2: In a classification task, not only the performance of the classification, but also the cases of misclassification need to be analyzed.

Response 2: We agree that misclassification rates are important. In the revised manuscript, we report accuracy, which is calculated as: 1 minus the misclassification rate. Given that we have an overall accuracy of 0.65, this means we have a misclassification rate of 0.35. We now clarify that point in the manuscript.

Comment 3: The details of iterations need to made clear. The number of features, features selected (even though frequently selected features have been shown in the manuscript), performance index values after each iteration, the convergence of the learning need to be shown.

Response 3: We agree and now include a new Table 2 that details the features and performance values for each iteration.

Comment 4: Word usage: it is better to use “machine learning” to replace “deep learning”.

Response 4: We have fixed the term.